# Pseudorandom Noise Forced Oscillation Technique to Assess Lung Function in Prematurely Born Children

**DOI:** 10.3390/children9081267

**Published:** 2022-08-22

**Authors:** Shannon Gunawardana, Christopher Harris, Anne Greenough

**Affiliations:** 1Department of Children’s and Women’s Health, School of Life Course Sciences, Faculty of Life Sciences and Medicine, King’s College London, London SE5 9RS, UK; 2NIHR Biomedical Research Centre based at Guy’s and St Thomas’ NHS Foundation Trust and King’s College London, London SE1 9RT, UK

**Keywords:** forced oscillation technique, preterm, follow-up

## Abstract

The forced oscillation technique (FOT) is a non-volitional assessment that is used during tidal breathing. A variant of FOT uses a pseudorandom noise (PRN) signal which we postulated might have utility in assessing lung function in prematurely born children. We, therefore, undertook a systematic review to evaluate the evidence regarding PRN FOT. A comprehensive search of the literature was conducted by using the following databases: Medline, Embase, Web of Science and CINAHL. Observational studies, case series/reports and randomized-controlled trials were eligible for inclusion. Article abstracts and full texts were screened independently by two reviewers, with disagreements resolved by discussion or a third reviewer if necessary. Five studies were included (n = 587 preterm children). Three compared PRN FOT with spirometry, and two compare it to the interrupter technique. Most studies failed to report comprehensive methodology of the frequency spectra used to generate the PRN signal. There was evidence that poorer lung function, as assessed by PRN FOT, was associated with a greater burden of respiratory symptoms, but there was insufficient evidence to determine whether PRN FOT performed better than other lung-function tests. Detailed methodological documentation, in accordance with ERS guidance, is needed to assess the benefits of PRN FOT prior to routine clinical incorporation to assess prematurely born children.

## 1. Introduction

Global estimates suggest that prematurity accounts for 10.6% of live births [1]. The UK EPICure studies and the Australian Victorian Infant Collaborative Study Group have reported reductions in prematurity-associated mortality over the last three decades [2,3]. Bronchopulmonary dysplasia (BPD), defined as a supplemental oxygen requirement for at least 28 days after birth [4], affects up to 75% of infants born at less than 28 weeks of gestational age (GA) [5]. Improvements in perinatal care, including greater use of antenatal steroids and postnatal surfactant, have been associated with improved neonatal outcomes. Nevertheless, prematurely born children still have impaired lung function as compared to their term-born peers, with airway obstruction being more severe in those who had BPD [6,7].

Given the growing population of prematurely born children with obstructive airway disease, assessing their lung function is of importance. Dubois et al. first described the forced oscillation technique (FOT) that superimposes sinusoidal pressure oscillations on tidal breathing to evaluate respiratory mechanics: the resistance (R) and reactance (X) of airways [8]. FOT has benefits over conventional lung-function testing in that it is a non-volitional effort-independent test which does not require complex respiratory maneuvers [9]. FOT is an umbrella term for different methods in which the oscillation signals can be mono- or multifrequency and use time-discrete impulses (impulse oscillometry, IOS) or continuous multifrequency sinusoidal waves (pseudorandom noise, PRN) [10]. This systematic review aimed to synthesize and evaluate the evidence regarding PRN FOT compared to other lung-function tests in prematurely born children.

## 2. Materials and Methods

This systematic review was conducted in accordance with the Preferred Reporting Items for Systematic Reviews and Meta-Analyses (PRISMA) guidelines and was registered in the International Prospective Register of Systematic Reviews, PROSPERO (CRD42022330039). The online search was carried out on the following databases: Medline via PubMed (1948–5 May 2022); Embase via Ovid SP (1947–5 May 2022); and the Cumulative Index to Nursing and Allied Health Literature (CINHL) via EBSCO host (1981–5 May 2022) and the Web of Science (1900–5 May 2022). The following terms were searched: (oscillometr * OR FOT OR ‘forced oscillation technique’ OR ‘oscillatory mechanics’), (lung * OR pulmonary OR resp * OR breath *), (child * OR adolesc * OR pediatric OR paediatric OR infan * OR teenage *), and (preterm OR pre-term OR ‘pre term’ OR premature OR pre-mature), combined with Boolean operators. Searches were performed without limitation on publication date or language.

The inclusion criteria were as follows: (1) studies that compared PRN FOT to other lung-function tests, (2) participants were children and young people less than 18 years of age and (3) studies that reported respiratory resistance (R) and reactance (X). Observational studies, case series and reports, and randomized-controlled trials were eligible for inclusion. Reviews and editorials were excluded. Other exclusion criteria were (1) conference abstracts, (2) full text in English unavailable, (3) patient data unavailable and (4) duplicated data from the same cohort of patients. S.G. carried out the preliminary search and de-duplication. Article abstracts and full texts were screened independently by two reviewers (S.G. and C.H.), with any disagreements resolved by discussion and a third reviewer (A.G.) if necessary (Figure 1 shows the PRISMA flowchart).

Data extraction was conducted by the primary reviewer (S.G.) of study characteristics, patient characteristics, outcome measures of oscillometry and other lung-function results. Data collection was completed by using a prewritten proforma. The Newcastle–Ottawa Scale was used to assess articles for bias, using a prewritten proforma that was reviewed by all authors prior to use [11].

A meta-analysis was not considered appropriate for this body of the literature because of the heterogeneity of comparator lung-function tests and the paucity of PRN FOT data. Instead, a narrative synthesis was performed to synthesize the findings of the studies. A preliminary synthesis was taken in the form of searching included studies and presenting characteristics and findings in a tabular form. Then the results were discussed again between reviewers and structured into themes. The following framework was used: methodological documentation, characteristics of cohorts, comparison of results with those of other lung-function tests and correlation with symptoms.

## 3. Results

Five studies were included in this review (Table 1).

### 3.1. Device Methodology

The spectrum of frequencies and devices used to generate PRN FOT varied between groups. Accorsi et al. employed a custom-made equipment of a wave-tube and loudspeaker that used a signal of multiple frequencies from 6 to 32 Hz [12]. Verheggen et al. used the i2M forced oscillation system, with a signal containing multiple frequencies between 2 and 48 Hz [13]. Vrijlandt et al. also used the i2M device, with a frequency spectrum of 4 to 48 Hz [14]. The other two studies used the i2M device, but the researchers did not report the frequency spectrum used [15,16].

### 3.2. Characteristics of Cohorts Studied

Most of the studies investigated children born extremely preterm (less than 28 weeks of GA) and/or very preterm (28 to less than 32 weeks of GA; see Table 1) [14,15,16,17,18]. Those of the BPD preterm group were of lower gestational age than those of the non-BPD preterm group in the Verheggen et al. (26.1 versus 29.4 weeks, *p* < 0.001) and Vrijlandt et al. (28 versus 29 weeks, *p* = 0.03) studies [13,14]. Verheggen et al. accounted for this in a stepwise multiple linear regression. Accorsi et al. were the only ones to investigate the moderate-to-late-preterm children (32 to less than 37 weeks of GA) [12,18]. Two groups studied preschool-aged children [14,16], two studied primary-school-aged children [15,17] and one studied secondary-school-aged children [14].

### 3.3. PRN FOT Compared to Other Lung-Function Tests

The detection of lung-function differences between term and preterm cohorts by PRN FOT compared to other lung-function tests was variable in the included studies (Table 2) [12,14,15,16,17].

Three studies compared PRN FOT against spirometry [12,15,17]. Simpson et al. studied 9-to-11-year-old children born at term and less than or equal to 32 weeks of gestational age [15]. They found that the PRN FOT results were significantly reduced (X8, *p* < 0.05) or significantly raised (Fres, *p* < 0.05, AX *p* < 0.001) in the preterm children compared to the term controls. The spirometry results of FEV1 (*p* < 0.001), FEV1/FVC (*p* < 0.001) and FEF25-75 (*p* < 0.001) were reduced in the preterm children. Verheggen et al. found that X8 (*p* < 0.02) could differentiate BPD from non-BPD children within a preterm cohort of 4-to-8-year-old children, but they did not detect significant differences in any other FOT or spirometry results [17]. Accorsi et al., in 11-to-14-year-old preterm and term children, found no significant differences in PRN FOT or spirometry results [12]. They did, however, note that the intra-breath oscillometry parameters of ‘change in resistance’ (∆R) and ‘reactance at end-inspiration’ (ReI) could differentiate the groups. Lombardi et al. showed that neither PRN FOT nor the interrupter technique demonstrated significant differences in the results of BPD and non-BPD five-year-old children born between 22 and 31 weeks of gestational age. Vrijlandt et al. found that the PRN FOT parameters of Fres and X4-24 significantly differed between BPD and non-BPD preterm children [14]. No significant difference between groups was detected by the results from the interrupter technique.

### 3.4. Correlation with Symptoms and Hospitalization

Lombardi et al. noted a significant association between wheeze with R8 (*p* = 0.04) and X8 (*p* = 0.04), but no significant association was noted with the interrupter technique Rint results (*p* = 0.14) [16]. Verhegen et al. also reported a significant correlation between wheeze and AX (*p* = 0.0009) and X8 (*p* = 0.03) amongst ex-BPD prematurely born children [17]. Simpson et al. found a correlation between respiratory symptom in the past three months and AX (*p* = 0.036) and X8 (*p* = 0.017) [15]. Vrijlandt et al. showed a higher Fres in prematurely born children who had been hospitalized with respiratory syncytial virus (RSV) than those without (*p* = 0.001) [14].

## 4. Discussion

This systematic review has demonstrated a paucity of studies that compare PRN FOT with other lung-function tests in prematurely born children. Some results suggest that PRN FOT can detect lung-function differences between premature and term-born children and between ex-BPD and non-BPD premature children, with correlation to respiratory symptoms. There is, however, variability in the results of the different studies, and this may relate to the different age groups studied and the possible effects of ‘catch up lung function’ in preterm children.

In most studies, the PRN FOT methods, however, were not comprehensively reported and lacked descriptions of frequency spectra used to generate composite sinusoidal waves. Three studies compared PRN FOT to spirometry, and two studies compared PRN FOT to the interrupter technique. PRN FOT was well-tolerated in children as young as preschool age, and more 4-to-8-year-old children performed acceptable FOT (99%) than spirometry measurements (62%) [13]. Although spirometry is often the routine clinical lung-function test, it can be challenging in preschool-aged children and yield unacceptable, invalid results [19]. Thus, FOT PRN might have a complimentary role in some patient cohorts.

PRN FOT was found to be superior to spirometry and the interrupter technique in detecting lung-function differences between BPD and non-BPD groups in some studies. Simpson et al. found significantly poorer lung function in preterm than term children, as assessed by spirometry results (FEV1: −0.72 versus 0.04, *p* < 0.001; FEV1/FVC: −1.25 versus −0.27, *p* < 0.001; FEF25-75: −1.46 versus −0.42, *p* < 0.001) and PRN FOT measures (X8: −0.43 versus 0.14, *p* < 0.05; AX: 0.29 versus −0.44, *p* < 0.001; Fres: 0.64 versus 0.18, *p* < 0.05) [15]. Similarly, reactance (X8), resonant frequency (Fres) and mean reactance (X4-24) were significantly poorer in BPD than non-BPD preterm children [14,17]. Vrijland et al. found that Rint, as measured by the interrupter technique, was similar in BPD and non-BPD groups, whereas PRN measures of Fres (26.8 versus 22.7, *p* = 0.001) and X4-24 (−3.0 versus 1.95, *p* = 0.008) were higher and lower, respectively, in BPD as compared to non-BPD prematurely born children [14,17]. Similarly, Verheggen et al. detected lower FOT X8 values (−1.48 vs. −0.89, *p* < 0.02) in BPD as compared to non-BPD preterm children, but they did not detect significant differences in any of the spirometry results. There is evidence that, in asthmatic children, oscillometry is more sensitive than spirometry to peripheral airway obstruction [20]. A significant association was found between PRN FOT results regarding RSV hospitalization and wheeze [14,17].

At low frequencies, resistance (R) is higher in distal airway obstruction, and reactance (X) is more negative in restriction or hyperinflation, with AX a marker of total respiratory reactance up to the resonant frequency (Fres) [10]. In the included studies, reactance (X) parameters, rather than resistance (R), detected lung-function differences between preterm and term children, and BPD and non-BPD preterm children [14,15,17]. This differs from IOS studies in which resistance (R) parameters better differentiate preterm and term children [21] and individuals with or without BPD [22]. A study that compared IOS with PRN FOT in adults with a variety of lung disorders found higher measures of R with IOS than FOT, especially at lower frequencies [23]. The mechanism of oscillation, pulses of IOS or continuous multifrequency sinusoidal waves of PRN, likely accounts for which FOT results might better detect lung-function differences in prematurely born children. The IOS signal differs from PRN in that signals are composed of harmonics of frequencies at multiples of 5 Hz, which might cause interference in impedance measures [9].

Oostveen et al. compared the i2M device with custom-made PRN devices and IOS in a large multicenter study of healthy adults [24]. They found that X at frequencies ≤ 14 Hz and R at all frequencies except 20 and 25 Hz were comparable across devices. The authors postulated that the observed frequency-specific variation could be due to population differences. They advocated for the use of X and R at low frequencies, along with AX and Fres, to detect lung-function abnormalities with oscillometric devices in adult populations. Another study compared IOS and commercial and custom-made PRN devices against test standards of known resistance and reactance in 12 healthy adults [25]. This group found that the devices (WIMR, tremoFlo, Resmon pro and IOS) gave comparable results to in vitro test standards of percentage predicted R (98.0, 100.0, 99.0, 100.0) and percentage predicted X (93.9, 104.0, 101.3); however, IOS could not measure X during the in vitro test. They noted significant variation in reactance (−0.76, −0.91, −0.69 and −0.98 cmH20 s/L) between all devices and in resistance of PRN devices (2.54, 2.60 and 2.50 cmH20 s/L) compared to IOS (3.77 cmH20 s/L). The authors suggest that the difference between in vitro and in vivo results could be due to breathing patterns and volume changes, which were not accounted for by the validation procedures. These data suggest the need for similar comparative studies of oscillometric devices in prematurely born children and the importance of human controls during routine use. It also suggests that oscillometric measures between FOT devices might not be comparable. In 2020, the European Respiratory Society (ERS) taskforce released technical standards for FOT, including the recommendation of a coefficient of variability (CoV) ≤ 15% over three repeat measures in pediatric testing [9]. The aim of this document was to standardize oscillometry measures through technical, hardware, software and patient factors.

Two of the included studies used PRN FOT to assess reversibility of airway obstruction by bronchodilator challenge [14,16]. Vrijlandt et al. did not detect a significant difference in the change in resistance at 6 Hz following bronchodilator challenge between the BPD (∆R6: 22.7% change in resistance) and non-BPD (∆R6: 18.0% change in resistance) cohorts [14]. Lombardi et al. found that a greater proportion of children had a positive bronchodilator response when assessed by the interrupter technique compared to FOT (18.4% Rint versus 9.9% AX) [16]. They defined a ‘significant’ bronchodilator response with the following cutoffs: R8 (−1.88 z-scores), X8 (+2.48 z-scores) and AX (+2.04 z-scores) [26]. Earlier this year, the ERS taskforce released advice for ‘significant’ bronchodilator response cutoff in children: 40% and 50% for R5 and X5, respectively [27]. They proposed those as a robust bronchodilator response cutoff across oscillometric devices, comparable to the 12% change in FEV1 advocated by the joint American Thoracic Society (ATS)-ERS taskforce [28]. We suggest that comparative studies that assess reversibility of airway obstruction with spirometry and PRN FOT evaluate those proposed cutoffs.

Another important consideration when interpreting studies is the variable reference ranges used. Accorsi et al. reported raw values, which should be interpreted cautiously given that height and sex have been shown to be independent determinants of R, X and AX [26]. The other four studies used three different reference ranges to calculate z-scores [26,29,30]. This could account for the large variability in z-scores of X8 (−0.28, −0.43, −1.25), AX (0.29, 0.29, 1.05) and R8 (−0.03, 0.33, 0.54) across preterm cohorts (Table 3).

The findings of Vrijlandt et al. were not tabulated, as they did not report z-scores for X8, AX or R8. Differences in characteristics such as sex distribution and gestational age could also account for this variability in oscillometry measures (Table 1). Prior to incorporation of PRN FOT into routine clinical practice, more data are required, using multiple FOT devices to create robust multi-ethnic reference ranges that follow the ERS technical standards [9].

Given the association between respiratory symptoms and FOT PRN results [14,15,16,17], there is scope for FOT to be used in wider settings, including in the emergency department and home monitoring for patients with chronic respiratory conditions. Indeed, airwave oscillometry (AOS), a type of FOT that uses a vibrating mesh to generate PRN, was used in a feasibility study to assess lung function in the pediatric emergency department [31]. Another group used an FOT device of single-frequency FOT to enable unsupervised home monitoring of lung function in adults with COPD [32]. They found that the day-to-day variability of FOT measurements were similar to that of supervised laboratory recordings.

## 5. Conclusions

PRN FOT can determine lung-function differences between preterm and term-born children at follow-up. More work is required to standardize different PRN FOT devices and, more broadly, FOT devices within this population. As prematurity-associated mortality improves, these children will make up an increasing proportion of pediatric respiratory-clinic patients. Prior to routine clinical incorporation, robust multi-ethnic reference equations are required, which include preterm-born children.

## Figures and Tables

**Figure 1 children-09-01267-f001:**
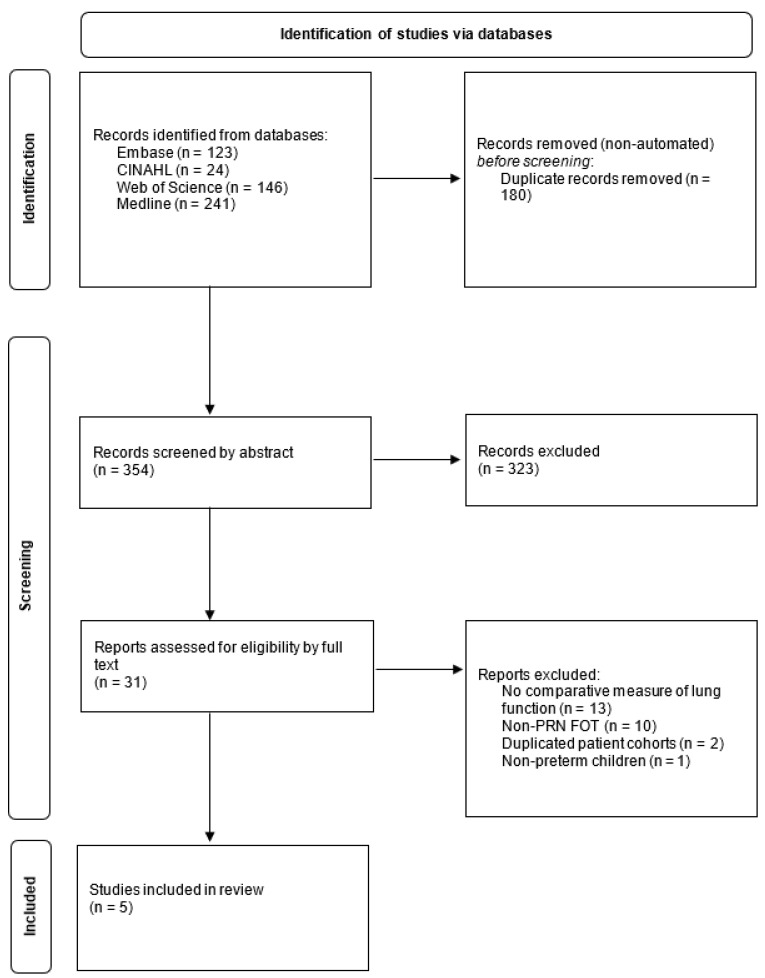
PRISMA flowchart; PRISMA: Preferred Reporting Items for Systematic Reviews and Meta-Analyses.

**Table 1 children-09-01267-t001:** Details of the studies included in the systematic review.

Author	Lung-Function Measures	Study Groups	Preterm (*n*)	Female (%)	Age (years)	GA (Weeks)	PRN FOT	Spirometry	Intra-Breath Oscillometry	Interrupter Technique (Rint)
Accorsi et al. [12]	PRN FOT; Spirometry; Intra-breath FOT	Term; preterm	35	51	12.9 (12.8–13.4)	33.5 +/− 1.5	Lower X6, X10 and Fres in preterm, not statistically significant	FVC, FEV1, FEV1/FVC and FEF25-75 all lower in preterm vs. term, not statistically significant	Lower change in resistance (0.06 vs. 0.46, *p* = 0.003), reactance at end-inspiration (−0.28 vs. −0.06, *p* = 0.027) in preterm	N/A
Lombardi et al.	PRN FOT; interrupter technique	BPD preterm; non-BPD preterm	194	46	5.2 (4.5–6.3) *	28 (25–31) *	Lower R8, X8 and AX in BPD vs. non-BPD groups, not statistically significant	N/A	N/A	Lower Rint score in BPD vs. non-BPD groups, not statistically significant
Simpson et al.	PRN FOT; spirometry; DLCO; multiple breath washout	Term: BPD preterm; non-BPD preterm	163	39	10.9 +/− 0.6	28.5 (25.0–29.6)	Lower X8 (−0.43 vs. 0.14, *p* < 0.05), higher AX (0.29 vs. −0.44, *p* < 0.001) and higher Fres (0.64 vs. −0.18 *p* < 0.05) in preterm vs. term	Lower FEV1 (−0.72 vs. 0.04, *p* < 0.001), FEV1/FVC (−1.25 vs. −0.27, *p* < 0.001), FEF25-75 (−1.46 vs. −0.42, *p* < 0.001) in preterm vs. term	N/A	N/A
Verheggen et al.	PRN FOT; spirometry	Term; BPD preterm; non-BPD preterm	118	40	BPD: 5.8 (4.4–7.3) **; Non-BPD: 6.0 (4.6–7.8) **	BPD: 26.1 (24.2–30.2) **; non-BPD: 29.4 (27.7–30.5)	Lower X8 in BPD vs. non-BPD preterm groups (−1.48 vs. −0.89, *p* < 0.02)	Lower FEV1 and FEV1/FVC in BPD vs. non-BPD groups, not statistically significant	N/A	N/A
Vrijlandt et al.	PRN FOT: interrupter technique	Term; BPD preterm; non-BPD preterm	77	48	BPD: 4.7 +/− 0.8; non-BPD: 4.8 +/− 0.8	BPD: 28 +/− 2 non-BPD: 29 +/− 2	Higher Fres (26.8 vs. 22.7, *p* = 0.001) and lower X4-24 (−3.0 vs. −1.95, *p* = 0.008) in BPD vs. non-BPD preterm groups	N/A	N/A	No significant differences between the groups

Data expressed as mean +/− SD or median (IQR), unless otherwise stated. * Data presented as median (range). ** Data presented as median (10th–90th centiles). DLCO, diffusion capacity of lung for carbon monoxide.

**Table 2 children-09-01267-t002:** Risk of bias evaluation—Newcastle–Ottawa Scale (11).

Author	Selection				Comparability	Outcome			Total
	Representative of the Exposed Cohort	Selection of the Non-Exposed Cohort	Ascertainment of Exposure	Demonstration That Outcome of Interest Was Not Present at Start of Study	Comparability of Cohorts on the Basis of the Design or Analysis	Assessment of Outcome	Was Follow-Up Long Enough for Outcomes to Occur?	Adequacy of Follow-Up of Cohorts	
Accorsi	0	1	1	0	1	1	1	0	5
Lombardi	1	1	1	0	1	1	1	1	7
Simpson	1	1	1	0	2	1	1	0	7
Verheggen	0	1	1	0	2	1	1	1	7
Vrijlandt	1	1	1	0	1	1	1	0	6

**Table 3 children-09-01267-t003:** Comparison of average z-scores between preterm cohorts.

Author	X8	R8	AX
Lombardi	−0.28	−0.03	0.29
Simpson	−0.43	0.33	0.29
Verheggen	−1.25	0.54	1.05

## Data Availability

All data generated or analyzed during this study are included in this article. Further enquiries can be directed to the corresponding author.

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
