# Peer review of "Pseudorandom Noise Forced Oscillation Technique to Assess Lung Function in Prematurely Born Children"

_children, 2022, doi:10.3390/children9081267_

Round 1

Reviewer 1 Report

1.       The effects of “catch-up lung function” in preterm children might be responsible for variable results across the studies. The authors should discuss the impact of age on the variable results across the studies.

2.       Discussion should be more precise and relevant to the study objective.

3.       There is no gold standard test ( either spirometry/interrupter technique) to identify differences in lung function between premature and term-born children. The authors' comment “ PRN FOT can detect” (line no 187) is not a valid conclusion.

Similarly, “PRN FOT found to be superior to spirometry” is also not a correct statement.

4.       The sentence is not a correction on many occasions e.g.

i.                     “reactance parameters of X8” (Line no 205). “X” itself stands for reactance.

ii.                   “preterm studies” (line no 215)

5.        The conclusion is not based on the review and should include guidance for further research.

Author Response

  1. The effects of “catch-up lung function” in preterm children might be responsible for variable results across the studies. The authors should discuss the impact of age on the variable results across the studies.

RESPONSE: We have now included this in the discussion

  1. Discussion should be more precise and relevant to the study objective.

RESPONSE: Apologies but do feel that our discussion was precise and relevant to the study objective which was to synthesise and evaluate the evidence for pseudorandom noise forced Oscillation Technique and compare the results to those of other lung function tests. In the discussion we focused the results of PRN FOT to other lung function tests, including following bronchodilator challenge and the correlation with respiratory symptoms. In addition, we commented on the limitation of the variable and the limited reference ranges used, an area which needed further research.

  1. There is no gold standard test ( either spirometry/interrupter technique) to identify differences in lung function between premature and term-born children. The authors' comment “ PRN FOT can detect” (line no 187) is not a valid conclusion.

Similarly, “PRN FOT found to be superior to spirometry” is also not a correct statement.

RESPONSE: We have now reworded our manuscript to say in certain studies PRN FOT can detect ----, but we did not state that spirometry was the gold standard rather that it was often the most commonly used respiratory function test in children.  

  1. The sentence is not a correction on many occasions e.g.
  2. “reactance parameters of X8” (Line no 205). “X” itself stands for reactance.
  3. “preterm studies” (line no 215)

RESPONSE: We have corrected these sentences

  1. The conclusion is not based on the review and should include guidance for further research.

RESPONSE: Apologies, we disagree with the reviewer. Our conclusion stated that “PRN FOT can  determine lung function differences between preterm and term born children at follow up. More work is required to standardise different PRN FOT devices and more broadly FOT devices, within this population. As prematurity-associated mortality improves, these children will make up an increasing proportion of paediatric respiratory clinic patients. Prior to routine clinical incorporation, robust multi-ethnic reference equations are required, which include preterm born children.”

As stated the conclusions are based on the review and we have included guidance for further research that is to standardize different PRN FOT devices and establish multi-ethnic reference ranges

Reviewer 2 Report

The authors have written a great review entitled " Pseudorandom Noise Forced Oscillation Technique to Assess Lung Function in Prematurely Born Children and Young People." Authors’ efforts deserve recognition; however, this review only included 5 studies of preterm infants but no young people. If there is no study of young people, the reason should be explained clearly or modify the title to exclude young people.

Author Response

The authors have written a great review entitled " Pseudorandom Noise Forced Oscillation Technique to Assess Lung Function in Prematurely Born Children and Young People." Authors’ efforts deserve recognition; however, this review only included 5 studies of preterm infants but no young people. If there is no study of young people, the reason should be explained clearly or modify the title to exclude young people.

RESPONSE: We thank the reviewer for their kind comments and have modified the title as suggested